# Effect of glucagon-like peptide-1 receptor agonists on glycemic control, and weight reduction in adults: A multivariate meta-analysis

Tzu-Lin Yeh[1,2], Ming-Chieh Tsai[2,3,4], Wen-Hsuan Tsai[4], Yu-Kang Tu[2], Kuo-Liong Chien[2,5]*

1 Department of Family Medicine, Hsinchu MacKay Memorial Hospital, Hsinchu, Taiwan, 2 Institute of Epidemiology and Preventive Medicine, College of Public Health, National Taiwan University, Taipei, Taiwan, 3 Department of Medicine, MacKay Memorial College, New Taipei City, Taiwan, 4 Division of Endocrinology, Department of Internal Medicine, MacKay Memorial Hospital, New Taipei City, Taiwan, 5 Department of Internal Medicine, National Taiwan University Hospital, Taipei, Taiwan

* klchien@ntu.edu.tw

**Data Availability Statement:** All relevant data are within the paper and its Supporting Information files.

## Abstract

### Aims

To explore the effect of glucagon-like peptide-1 receptor agonist (GLP-1 RAs) on glycemic control and weight reduction in adults.

### Methods

Databases were searched from August 2021 to March 2022. Data were analyzed using mean difference (MD) values with 95% confidence intervals (CIs). Both random-and fixed-effect models were employed. Heterogeneity was explored using pre-specified subgroup analyses and meta-regression. Structural equation modeling fitting was used for the multivariate meta-analysis.

### Results

A total of 31 double-blind randomized controlled trials with 22,948 participants were included in the meta-analysis. The MD and 95% CI of the pooled GLP1-RA-induced change in the glycated hemoglobin level was -0.78% (-0.97%, -0.60%) in the random-effects model and -0.45% (-0.47%, -0.44%) in the fixed-effect model, with a high heterogeneity ($I^2$ = 97%). The pooled body weight reduction was -4.05 kg (-5.02 kg, -3.09 kg) in the random-effects model and -2.04 kg (-2.16 kg, -1.92 kg) in the fixed-effect model ($I^2$ = 98%). The standardized pooled correlation coefficient between HbA1c levels and body weight was -0.42. A negative correlation between glycemic control and weight reduction was obtained.

### Conclusion

Long-acting GLP-1 RAs significantly reduced the glycated hemoglobin level and body weight in adults.

**Funding:** The authors received no specific funding for this work.

**Competing interests:** The authors have declared that no competing interests exist.

## Introduction

Glucagon-like peptide 1, an incretin secreted from the gut, exerts metabolic effects through glucose-dependent stimulation of insulin secretion, delayed gastric emptying, inhibition of appetite, and increased natriuresis [1]. Glucagon-like peptide receptor agonists (GLP-1 RAs) have been used to treat patients with diabetes since 2007 [2]. and have been approved as anti-obesity drugs since 2014 [3]. However, long-acting GLP-1 RAs have attracted increasing interest due to their better efficacy in diabetes and obesity treatment [4]. Long-acting GLP-1 RA treatment was shown to be associated with a pooled glycated hemoglobin (HbA1c) reduction of 0.99% and a pooled body weight reduction of 2.69 kg (heterogeneity, approximately 90%) [5]. The high heterogeneity can be partially explained by differences in the underlying conditions of participants [6] and the GLP-1 RA interventions [7]. Moreover, participant age and the baseline glycemic level may interact with the results in children [6], indicating the existence of potential effect modifiers. However, further analysis to explore the high heterogeneity and potential effect modifiers in adults is lacking [8, 9].

Glycemic control is intertwined with the weight reduction caused by long-acting GLP-1 RAs through insulin resistance and metabolic changes [10]. Thus, these two outcomes of interest should not be independently estimated. However, previous randomized controlled trials (RCTs) rarely reported the correlation coefficients at the within-study level [11], and to the best of our knowledge, no correlation coefficient was reported in between-study-level meta-analysis [12, 13].

Thus, to explore the high heterogeneity and possible effect modifiers associated with these findings, we performed further univariate meta-analyses of the glycemic control and weight reduction caused by long-acting GLP-1 RAs in adults. Considering the correlation between these outcomes, our study used the structural equation modeling approach for multivariate meta-analysis to jointly estimate the effect sizes for glycemic control and weight reduction in one model and to investigate the associations between these two outcomes of long-acting GLP-1 RA treatment.

## Materials & methods

### Search strategy and selection criteria

We searched the Medline, Ovid EMBASE, Cochrane Library and ClinicalTrials.gov databases for relevant studies from August 2021 to March 2022 by using the following keywords: "Glucagon-Like Peptide 1" OR "GLP-1" OR "Placebo" OR "Body Weights" OR "Glucose" OR "Glycosylated Hemoglobin A" OR "Trials, Randomized Clinical." The PRISMA checklist and detailed search strategies are shown in Supplement and **S1 Table**. To enable a comprehensive search, we did not include limiting parameters for language, article type, year of publication, animal or human subjects, and age of participants.

We included all eligible publications that met the following inclusion criteria: (1) adult participants older than 18 years, either from the general population or including patients with a specific disease; (2) intervention with U.S. Food and Drug Administration approved long-acting GLP-1 RAs, including liraglutide, once-weekly exenatide, dulaglutide, albiglutide, and semaglutide, which were administered orally or subcutaneously, either in same or different doses; (3) comparison with a placebo; (4) glycemic or anthropometric changes as either primary or secondary outcome measures; (5) phase 3 or phase 4 randomized, double-blind, placebo-controlled trials without cross-over or open-label in any study period. We excluded articles that met the following criteria: (1) were duplicated publications or used duplicated populations, such as a post-hoc analysis of an included trial; (2) included participants with

other conditions that interfered with outcome assessment, such as pregnancy or weight reduction surgery; (3) assessed other active components in addition to GLP-1 RAs in the treatment arm; (4) performed active comparisons rather than comparisons with placebo; (4) used outcome measures that were not of our interest; (6) reported conference abstracts, review articles, or phase 1 or 2 RCTs. All included trials were assessed for bias using the Cochrane risk-of-bias tool 2.0 [14]. The details of the data extraction in our study were described in supplement (S1 File).

Data were analyzed using the mean difference (MD) with 95% confidence intervals (CIs) for continuous outcomes. For the univariate meta-analysis, we used the statistical software R, version 4.0.3, and the *meta* package. Both random- and fixed-effect models were employed using DerSimonian and Laird's method [15]. The results of the meta-analysis are presented in forest plots. Heterogeneity was quantified using the Cochran Q test and $I^2$ statistics [16]. Heterogeneity was explored in pre-specified subgroup analyses by participants' disease and intervention drugs. Potential effect modifiers were determined in meta-regression analysis. Publication bias was inspected using the symmetry of the funnel plot and Egger's test [17]. Contour-enhanced funnel plots to enhance the recognition of the causes of asymmetry and trim-and-fill analysis to estimate the effect size were performed if a bias existed. To ensure robustness, a further meta-analysis restricted to articles with a low risk of bias was performed. Since the correlation was not reported in each original study, we set the correlation coefficient between HbA1c level and body weight changes as 0.2, based on a reasonable assumption and previous literature [11]. We used the *metaSEM* package to fit the structural equation modeling using the maximum likelihood estimation in one step. Effect sizes and effect size variances were the essential arguments to be specified. The results of the multivariate meta-analysis model were visualized by plotting. To explore the direction of the pooled correlation coefficient, we further restricted the multivariate meta-analysis according to participant characteristics. Sensitivity analyses were performed by setting other correlation coefficients and restricting to studies with a low risk of bias.

## Results

### Description of studies and quality assessment

Thirty-one double-blind RCTs [18–47] were included in our meta-analysis (**Fig 1**). Seven of these were phase 4 trials [24, 25, 29, 35, 38, 42, 43]. The eligible participants ranged from non-diabetic overweight/obese general individuals to patients with schizophrenia [31], obstructive sleep apnea [20], or polycystic ovary syndrome [25]; patients with type 1 diabetes mellitus (DM) [24, 28, 33]; various groups of type 2 DM patients, including drug-naïve patients [19, 41], those treated with insulin [18, 26, 40, 46], those with chronic kidney disease [22, 36], and those with cardiovascular disease [27, 30, 34]. Liraglutide was the most commonly used GLP-1 RA, followed by subcutaneous semaglutide [21, 34, 40, 41, 44, 47], oral semaglutide [19, 30, 36, 46] and dulaglutide [27], once-weekly exenatide [31], and albiglutide [37]. A total of 23,061 participants (mean age, 54.1 years; 54.1% women; baseline body mass index (BMI), 33.7 kg/m²; baseline HbA1c, 7.8%; mean disease duration, 8.2 years; mean study period, 38.1 weeks) were included in the univariate meta-analysis. The baseline characteristics of the included studies are presented in **S2 Table**. In the GLP-1 RA arm, a total of 12,319 participants (mean age, 54.1 years; 54.4% women, mean baseline BMI, 33.9 kg/m²; mean baseline HbA1c level, 7.7%; mean follow-up duration, 38.1 weeks; mean duration of diabetes, 8.0 years) were included in the multivariate meta-analysis. Most of the included RCTs were assessed as showing high quality with a low risk of bias; only five trials [28, 31, 32, 35, 38] did not use the intention-to-treat analysis and were thus assessed as showing some concerns (**S3 Table**).

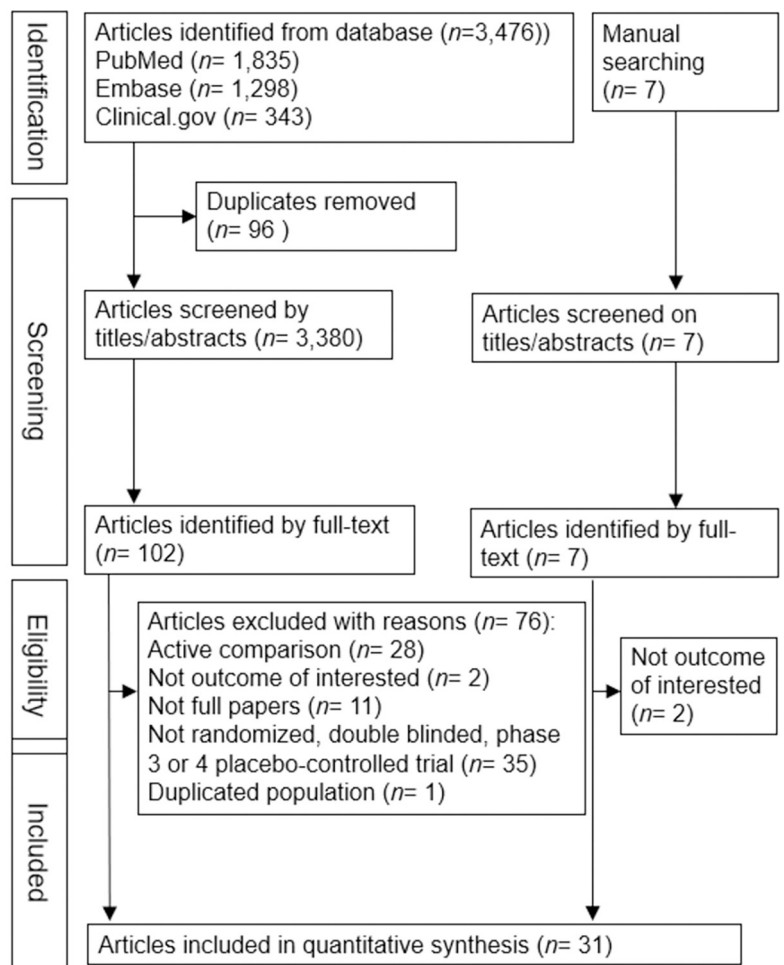

**Fig 1. Flowchart of the trial selection process.**

### Results of the univariate meta-analysis

The MD (95% CI) for the pooled HbA1c change caused by GLP-1 RAs was -0.78% (-0.97%, -0.60%) in the random-effect model and -0.45% (-0.47%, -0.44%) in the fixed-effect model, with a high heterogeneity ($I^2$ = 97%). Forest plots are shown in **S1 Fig**. Subgroup analysis based on participant characteristics showed that the MD (95% CI) for the pooled HbA1c change was -0.99% ([-1.17%, -0.82%], $I^2$ = 94%) in type 2 DM patients, -0.27% ([-0.31%, -0.24%], $I^2$ = 39%) in participants with overweight/obesity, and -0.18% ([-0.35%, -0.01%], $I^2$ = 0%) in type 1 DM patients (**S2 Fig**). The MD (95% CI) for the pooled HbA1c change was -1.06% ([-1.50%, -0.62%], $I^2$ = 99%) with subcutaneous semaglutide, -0.54% ([-0.76%, -0.33%], $I^2$ = 93%) with liraglutide, -0.94% ([-1.18%, -0.70%], $I^2$ = 89%) with oral semaglutide, and -0.82% ([-1.23%, -0.41%], $I^2$ = 80%) with dulaglutide/exenatide/albiglutide (**S3 Fig**). Meta-regression (**Table 1**) showed that the pooled HbA1c reduction significantly interacted with participants' baseline age ($p$ = 0.032), proportion of female participants ($p$ = 0.017), the baseline HbA1c level ($p$ = 0.018), and gastrointestinal side effect ($p$ = 0.002) but did not interact with baseline body weight or BMI level, duration of diabetes, follow-up period or insulin use. A one-year increase in the participants' age significantly decreased the pooled HbA1c change by 0.026%; a 1% increase in the proportion of female participants significantly increased the

**Table 1. Univariate meta-regression of the effects of glucagon-like peptide-1 receptor agonists on glycated hemoglobin levels and weight reduction.**

| | Glycated hemoglobin | | | Body weight | | |
|---|---|---|---|---|---|---|
| | MD (95% CI) | $p$ | $R^2$(%) | MD (95% CI) | $p$ | $R^2$(%) |
| Age (years) | -0.026 (-0.049, -0.002) | 0.032 | 17.2 | 0.093 (-0.016, 0.202) | 0.091 | 6.0 |
| Proportion of women (%) | 0.015 (0.003, 0.027) | 0.017 | 17.3 | -0.065 (-0.117, -0.013) | 0.016 | 18.0 |
| Baseline glycated hemoglobin (%) | -0.262 (-0.473, -0.050) | 0.018 | 25.1 | 1.130 (-0.132, 2.393) | 0.077 | 9.6 |
| Baseline body mass index (kg/m$^2$) | 0.021 (-0.045, 0.087) | 0.514 | 0 | -0.217 (-0.553, 0.120) | 0.197 | 3.5 |
| Baseline body weight (kg) | 0.010 (-0.013, 0.033) | 0.374 | 1.8 | -0.045 (-0.157, 0.068) | 0.424 | 0 |
| Duration of diabetes (years) | -0.004 (-0.035, 0.027) | 0.788 | 0 | 0.105 (-0.060, 0.269) | 0.203 | 3.7 |
| Study duration (weeks) | 0.002 (-0.007, 0.011) | 0.660 | 0 | -0.052 (-0.094, -0.010) | 0.018 | 16.2 |
| Gastrointestinal side effect (%) | 0.017 (0.007, 0.027) | 0.002 | 35.1 | -0.059 (-0.110, -0.008) | 0.025 | 15.1 |
| Insulin use (%) | 0.000 (-0.005, 0.005) | 0.863 | 0 | 0.015 (-0.011, 0.040) | 0.249 | 2.2 |

CI, confidence interval; MD, mean difference; $R^2$ (%), percentage of heterogeneity explained

pooled HbA1c change by 0.015%; a 1% increase in the baseline HbA1c level significantly reduced the pooled HbA1c level by 0.262%; and 1% increase in gastrointestinal side effect significantly increase the pooled HbA1c level by 0.017%. The funnel plot (**S4 Fig**) showed asymmetry, and publication bias was confirmed by Egger's test ($p$ = 0.004). To ensure that the bias did not contribute to the underlying differences between studies, we further omitted outliers [31], participants without diabetes [31, 44], and participants without type 2 diabetes [24, 28, 33], and the Egger's test still showed significant bias (**S5 Fig**). In the assessment of the reasons for this asymmetry, the contour-enhanced funnel plot indicated that studies with a positive MD with any $p$ value were not found (**Fig 2**). Therefore, we conducted a trim-and-fill analysis to examine the influence of publication bias. The trim-and-fill analysis showed 10 unpublished studies. Considering these unpublished studies, the MD (95% CI) for the pooled HbA1c change was -0.45% (-0.58%, -0.31%), which was also similar to the results of the fixed-effect model. Further analysis by restricting articles with low bias showed robust results (-0.78% [-0.97%, -0.59%], $I^2$ = 98%) (**S6 Fig**).

The pooled body weight reduction caused by GLP1-RA was -4.05 kg (-5.02 kg, -3.09 kg) in the random-effects model and -2.04 kg (-2.16 kg, -1.92 kg) in the fixed-effect model ($I^2$ = 98%). Forest plots are shown in **S1 Fig**. Subgroup analysis by participants' characteristics (**S2 Fig**) showed that the MD (95% CI) for the pooled body weight change was -3.14% ([-3.84%, -2.44%], $I^2$ = 97%) in type 2 DM patients, -5.77% ([-8.35%, -3.20%], $I^2$ = 99%) in participants with overweight/obesity, and -4.15% ([-5.04%, -3.25%], $I^2$ = 0%) in type 1 DM patients. The MD (95% CI) for the pooled body weight change was -6.58% ([-9.24%, -3.92%], $I^2$ = 98%) with subcutaneous semaglutide, -3.87% ([-4.57%, -3.17%], $I^2$ = 73%) with liraglutide, -2.91% ([-3.48%, -2.34%], $I^2$ = 67%) with oral semaglutide, and -0.47 ([-1.50%, 0.56%], $I^2$ = 98%) with dulaglutide/exenatide/albiglutide (**S3 Fig**). Meta-regression (**Table 1**) showed that the pooled body weight reduction significantly interacted with the proportion of female participants ($p$ = 0.016), the follow-up period ($p$ = 0.018) and gastrointestinal side effect ($p$ = 0.025), and had a borderline interaction with participants' baseline age ($p$ = 0.091) and the participants' baseline HbA1c level ($p$ = 0.077), but did not interact with baseline BMI level or body weight, insulin use, or the duration of diabetes. A 1% increase in the proportion of female participants significantly decreased the pooled body weight reduction by 0.065 kg; a one-week increase in the treatment duration significantly decreased the pooled body weight reduction by 0.052 kg; a 1% increase in gastrointestinal side effect significantly decreased the pooled body weight reduction by 0.059 kg. Publication bias was present, and the funnel plot and Egger's test results

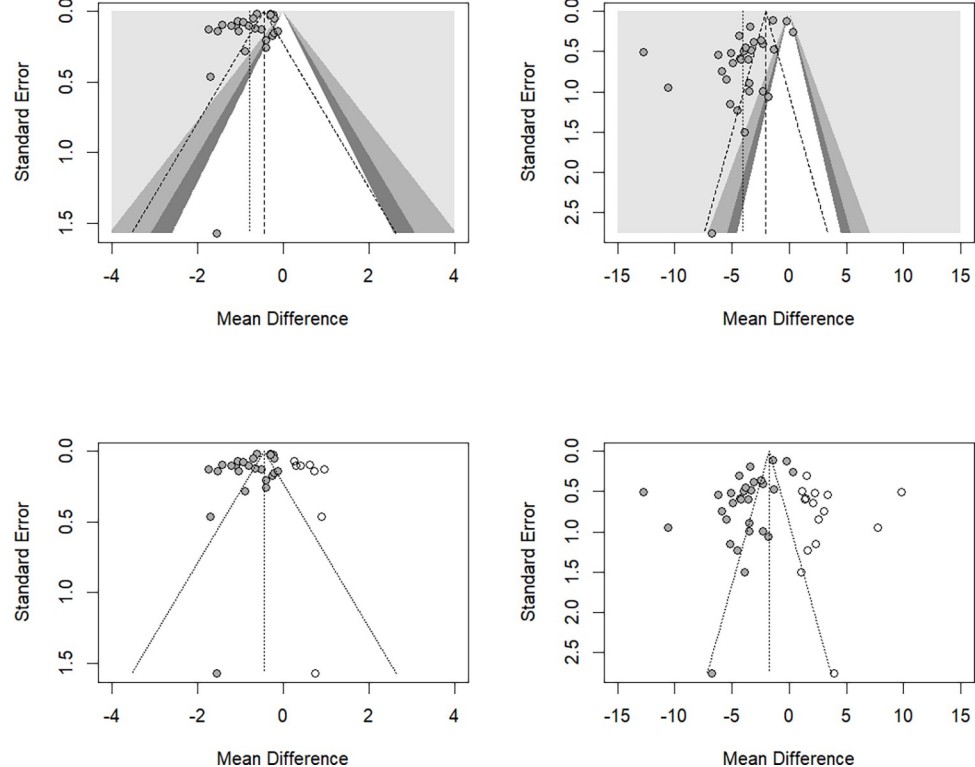

**Fig 2. Contour-enhanced and filled funnel plots of glycated hemoglobin level and body weight.** (upper left) Contour-enhanced funnel plot of the pooled glycated hemoglobin level after treatment with a glucagon-like peptide-1 receptor agonist. (lower left) Filled funnel plot of the pooled glycated hemoglobin level after treatment with a glucagon-like peptide-1 receptor agonist. (upper right) Contour-enhanced funnel plot of the pooled body weight after treatment with a glucagon-like peptide-1 receptor agonist. (lower right) Filled funnel plot of the pooled body weight after treatment with a glucagon-like peptide-1 receptor agonist.

(*p* value < 0.001) are shown in **S4 Fig**. After omitting outliers [45] and restricting the study to participants with diabetes or type 2 diabetes, Egger's test still showed significant bias (**S5 Fig**). The contour-enhanced funnel plot also showed the absence of studies with a positive MD, and publication bias existed (**Fig 2**). The trim-and-fill analysis revealed 15 unpublished studies (**Fig 2**). Considering these unpublished studies, the MD (95% CI) of the pooled body weight reduction was -1.76 kg (-2.63 kg, -0.88 kg), which was almost half of the current results and similar to the results obtained with the fixed-effect mode. Further analysis by restricting articles with low bias showed similar results: -4.29% ([-5.37%, -3.22%], $I^2$ = 98%) (**S6 Fig**).

## Results of the structural equation modeling multivariate meta-analysis

Maximum likelihood estimation worked well in the analysis. **Table 2** shows that the pooled HbA1c change induced by GLP1-RAs was -0.85% (95% CI [-1.03%, -0.66%], $I^2$ = 99%), and the pooled body weight change was -4.03 kg (95% CI [-5.11 kg, -2.95 kg], $I^2$ = 99%), which were similar to the results of the univariate meta-analysis. However, overall, the pooled between-study level correlation coefficient between HbA1c and body weight changes from baseline was -0.42, which was the opposite of the within-study level. To explore the negative correlation, we further restricted the multivariate analysis to participants with or without diabetes. The pooled HbA1c change by GLP1-RA was -0.96% (95% CI [-1.14%, -0.79%], $I^2$ = 96%), the pooled body weight change was -3.23 kg (95% CI [-3.86 kg, -2.59 kg], $I^2$ = 95%); the

**Table 2. The pooled results for glycated hemoglobin level and weight reduction on comparison of glucagon-like peptide-1 receptor agonist and placebo by using the structural equation modeling multivariate meta-analysis according to participant characteristics.**

| Setting $r = 0.2$ | All participants | | | Patients with diabetes | | | Patients without diabetes | | |
|---|---|---|---|---|---|---|---|---|---|
| | Estimates (95% CI) | $p$ | $I^2$ | Estimates (95% CI) | $p$ | $I^2$ | Estimates (95% CI) | $p$ | $I^2$ |
| Glycated hemoglobin (%) | -0.85 (-1.03, -0.66) | <0.001 | 99% | -0.96 (-1.14, -0.79) | <0.001 | 96% | -0.27 (-0.31, -0.23) | <0.001 | 13% |
| Body weight (kg) | -4.03 (-5.11, -2.95) | <0.001 | 99% | -3.23 (-3.86, -2.59) | <0.001 | 95% | -6.76 (-10.81, -2.72) | 0.001 | 99% |
| tao of glycated hemoglobin | 0.18 (0.07, 0.30) | <0.001 | | 0.13 (0.03, 0.23) | 0.008 | | 0 | 0.909 | |
| tao of body weight | 7.36 (3.10, 11.63) | <0.001 | | 1.77 (0.56, 2.99) | 0.004 | | 20.84 (-5.32, 47.00) | 0.118 | |
| Covariance | -0.48 (-1.02, 0.05) | <0.001 | | 0.15 (-0.12, 0.43) | 0.281 | | 0.04 (-0.14, 0.21) | 0.682 | |
| Standardized correlation coefficient | -0.42 | | | 0.32 | | | 0.81 | | |

CI, confidence interval; $r$, correlation coefficient between glycated hemoglobin and body weight changes within the study level; tao, the variance of effect measure

amount of between-study heterogeneity of body weight decreased from 7.36 to 1.77 and the 95% CI became narrower. The pooled correlation coefficient turned to a positive estimate of 0.32. There were only five studies focused on participants without diabetes. The pooled HbA1c change by GLP1-RA was -0.27% (95% CI [-0.31%, -0.23%], $I^2$ = 13%), the pooled body weight change was -6.76 kg (95% CI [-10.81 kg, -2.72 kg], $I^2$ = 99%); the amount of between-study heterogeneity of body weight was much increased to 20.84 with a wide 95% CI due to the limited included articles. However, the pooled correlation coefficient was positive of 0.81. The pooled results for all participants and the results restricted to patients with diabetes are shown in **Fig 3**. Sensitivity analyses of all participants and patients with diabetes by setting the correlation coefficient to 0.1 and 0.3 and restricting the study selection to studies with a low risk of bias

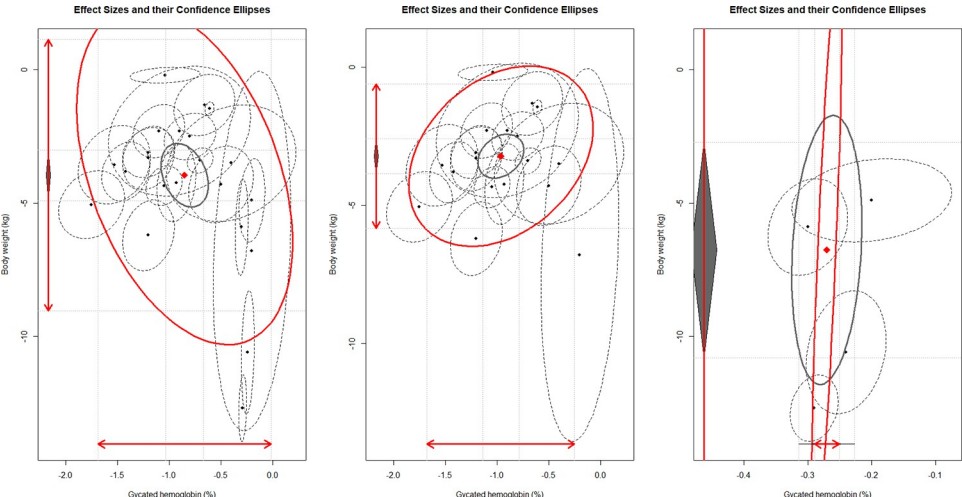

**Fig 3. The pooled effect sizes and their confidence ellipse for changes in glycated hemoglobin level and body weight in patients treated with a glucagon-like peptide-1 receptor agonist and placebo, in participants not restricted and restricted to diabetes.** (left) The pooled effect sizes and their confidence ellipse of the changes in the glycated hemoglobin level and body weight, in a comparison of glucagon-like peptide-1 receptor agonists and placebo in all participants. (middle) The pooled effect sizes and their confidence ellipse for changes in the glycated hemoglobin level and body weight, in a comparison of glucagon-like peptide-1 receptor agonists and placebo in patients with diabetes. (right) The pooled effect sizes and their confidence ellipse for changes in the glycated hemoglobin level and body weight, in a comparison of glucagon-like peptide-1 receptor agonists and placebo in patients without diabetes. x-axis: Effects of glucagon-like peptide-1 receptor agonists on glycated hemoglobin level; y-axis: effects of glucagon-like peptide-1 receptor agonists on body weight changes; black dots: individual studies; ellipses with dashed lines: 95% confidence interval; red diamond: pooled effect and 95% confidence interval; smaller gray ellipse: 95% confidence interval; larger red ellipse: 95% prediction interval.

yielded robust results (**S4 Table**). Further sensitivity analyses of participants without diabetes were not performed due to the limited included articles.

## Discussion

Our meta-analysis demonstrated that long-acting GLP-1 RAs significantly reduced HbA1c levels and body weight in adults. The high heterogeneity in our study might be attributed to the different GLP-1 RAs and the diverse populations ranging from non-diabetic overweight/obese participants to patients with diabetes complicated with end organ damage. The effects of GLP-1 RA were highly correlated with age, sex, baseline condition, treatment duration, and gastrointestinal side effect. For the correlation between glycemic control and weight reduction, the pooled effects were similar, since both effects were estimated independently. However, although the long-acting GLP-1 RA lowered the HbA1c more, it did not cause much decrease in the body weight in our included population. A positive association was found only in a specific condition.

Long-acting GLP-1 RAs showed better efficacy in weight reduction and glycemic control than short-acting GLP-1 RAs [5]. After the first wave of approvals for long-acting GLP-1 RAs from 2009 to 2014 [48–50], semaglutide was approved in 2017 [51], and the oral form of semaglutide was recently approved in 2020 [52]. Thus, meta-analyses published before 2015 [7, 12, 13, 53, 54] did not discuss all the currently available long-acting GLP-1 RAs, while more recent meta-analyses usually targeted semaglutide [8, 9] or focused on emerging outcomes such as cardiovascular or kidney disease [55, 56]. In contrast, our study aimed to investigate the glycemic control and weight reduction caused by long-acting GLP-1 RAs. Previous studies showed high heterogeneity ($I^2$ = 80%-90%) even for findings related to the same GLP-1 RAs [8, 9, 57]. All the potential effect modifiers in our study showed an opposite direction of interaction between glycemic control and weight reduction. Previous meta-analyses have rarely reported this topic and yielded inconsistent results [6, 57]; further studies are warranted to explore a potential effect modifier in the complex combinations between different interventions and target populations. Although unpublished studies (NCT01753362, NCT03480022, NCT02417142, NCT02473809, NCT04325581, NCT03048578, NCT01455441, NCT03466021, NCT04109547, NCT03811574, NCT03693430), withdrawn studies (NCT04057261, NCT02229240), terminated studies (NCT03279731, NCT01628445), studies with an unknown status (NCT01722240, NCT04046822, NCT01722240, NCT02016846, NCT04126603) or those on albiglutide, which was withdraw from the market, all possibly explained the publication bias, the pooled results for the significant HbA1c- and body weight-lowering effects remained robust.

According to a previous within-study-level study [11] and between-study-level network meta-analysis [58], GLP1 RAs showed a higher efficacy for glycemic control, and a compatible higher efficacy for weight reduction is expected. The ecological fallacy in our pooled negative associations between glycemic control and weight reduction may be partially explained by a publication bias, but was better explained by a higher coefficient of variation of GLP-1 RAs for reducing body weight than HbA1c levels and the underlying glucose level [4]. After removing these influential points, studies [20, 31, 44, 45] with prominent weight-reduction effects and modest effects on glycemic changes in non-diabetic participants yielded positive pooled results, supporting our explanation. The mechanism underlying the variable effects of GLP-1 RAs on body weight is not well understood. GLP-1 RAs decreased appetite through direct effects on the hypothalamus, neuronal activation in brain areas, reduced caloric intake, and interference of effective compensatory mechanisms counteracting weight loss [59–61].

Our study provided evidence that the actual effects of GLP-1 RAs on glycemic control and weight reduction were not as high as those reported in previous studies. Thus, a more conservative view of the current published results on GLP-1 RAs is recommended. Clinicians could expect positive associations between weight reduction and glycemic control in diabetes patients treated with GLP-1 RAs. However, marked weight loss in a non-diabetic patient in response to GLP-1 RA treatment did not indicate that clinicians could expect a corresponding glycemic improvement due to the results were interacted with the underlying glucose level.

To the best of our knowledge, our study is the first to consider the correlation between two dependent variables and estimate the relationship between the glycemic control and weight-reducing effects of GLP-1 RAs jointly with an unbiased methodology, a structural equation modeling approach for a multivariate meta-analysis. We comprehensively investigated heterogeneity, effect modifiers, and the reasons for and impact of publication bias. However, the study also had some limitations. First, confirmatory factor and mediation analyses were not performed. We contacted the original authors, but the lack of correlations among the observed variables in individual studies hindered further analysis. Second, our study focused on long-acting GLP1-RAs in comparison with placebo, and future studies should expand the scope to include short-acting GLP1-RAs and comparisons with active components or to sodium-glucose cotransporter-2 inhibitors. Third, the model could not estimate correlations between cardiovascular outcomes and glycemic control and/or weight reduction; the within-study correlation coefficients were not available for categorical variables.

In conclusion, long-acting GLP-1 RAs significantly lowered HbA1c levels and body weight in adults. However, the positive association between glycemic control and weight reduction was only observed in diabetic patients and in non-diabetic participants, but not in all participants with high heterogeneity treated with long-acting GLP-1 RAs.

## Supporting information

**S1 Checklist. PRISMA checklist 2020.**
(PDF)

**S1 File. Details of data extraction in the study.**
(DOCX)

**S1 Table. Search strategy.**
(PDF)

**S2 Table. Characteristics of included randomized double-blind placebo-controlled studies.**
(PDF)

**S3 Table. Summary of risk of bias assessment for included studies.**
(PDF)

**S4 Table. Sensitivity analysis of the pooled results for glycated hemoglobin level and weight reduction on comparison of glucagon-like peptide-1 receptor agonist and placebo by using the structural equation modeling multivariate meta-analysis according to participant characteristics.**
(PDF)

**S1 Fig. Forest plots of univariate meta-analysis.** CI, confidence interval; MD, mean difference; SE, standard error; TE, treatment effect.
(JPG)

**S2 Fig. Forest plots of univariate meta-analysis, subgroup by participants' characteristics.** CI, confidence interval; DM, diabetes mellitus; MD, mean difference; SE, standard error; TE, treatment effect.
(TIF)

**S3 Fig. Forest plots of univariate meta-analysis, subgroup by different glucagon-like peptide-1 receptor agonists.** CI, confidence interval; GLP1 RA, Glucagon-like peptide-1 receptor agonist; MD, mean difference; sc, subcutaneous; SE, standard error; TE, treatment effect. GLP1 = Others referred to Dulaglutide, once-weekly Exenatide and Albiglutide.
(TIF)

**S4 Fig. Funnel plots and the Egger's tests of the univariate meta-analysis.**
(TIF)

**S5 Fig. Funnel plots and the Egger's tests of the univariate meta-analysis in different conditions.** DM, diabetes mellitus; HbA1c, glycated hemoglobin.
(TIF)

**S6 Fig. Forest plot of pooled glycated hemoglobin level and body weight reduction, comparing Glucagon-like peptide-1 receptor agonist and placebo, by restricting articles with low bias.** CI, confidence interval; MD, mean difference; SE, standard error; TE, treatment effect.
(JPG)

## Acknowledgments

We would like to thank Editage for editing and proofreading this manuscript.

## Author Contributions

**Conceptualization:** Tzu-Lin Yeh, Yu-Kang Tu, Kuo-Liong Chien.

**Data curation:** Tzu-Lin Yeh, Ming-Chieh Tsai, Wen-Hsuan Tsai.

**Formal analysis:** Tzu-Lin Yeh.

**Methodology:** Tzu-Lin Yeh, Ming-Chieh Tsai, Wen-Hsuan Tsai, Yu-Kang Tu, Kuo-Liong Chien.

**Software:** Tzu-Lin Yeh.

**Supervision:** Yu-Kang Tu, Kuo-Liong Chien.

**Validation:** Tzu-Lin Yeh, Ming-Chieh Tsai.

**Writing – original draft:** Tzu-Lin Yeh.

**Writing – review & editing:** Tzu-Lin Yeh.

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
