## [Decision Letter · Decision Letter 0]

21 Oct 2022

PONE-D-22-25301Association between glycemic control and weight reduction of glucagon-like peptide-1 receptor agonists: a multivariate meta-analysisPLOS ONE

Dear Dr. Yeh,

Thank you for submitting your manuscript to PLOS ONE. After careful consideration, we feel that it has merit but does not fully meet PLOS ONE’s publication criteria as it currently stands. Therefore, we invite you to submit a revised version of the manuscript that addresses the points raised during the review process.

We look forward to receiving your revised manuscript.

Kind regards,

Ming-Chang Chiang

Academic Editor

PLOS ONE

Journal Requirements:

3. We note that this manuscript is a systematic review or meta-analysis; our author guidelines therefore require that you use PRISMA guidance to help improve reporting quality of this type of study. Please upload copies of the completed PRISMA checklist as Supporting Information with a file name “PRISMA checklist” and it should be uploaded separately.

Reviewers' comments:

Reviewer's Responses to Questions

**Comments to the Author**

1. Is the manuscript technically sound, and do the data support the conclusions?

Reviewer #1: Yes

Reviewer #2: Partly

Reviewer #3: Yes

2. Has the statistical analysis been performed appropriately and rigorously? 

Reviewer #1: Yes

Reviewer #2: Yes

Reviewer #3: I Don't Know

3. Have the authors made all data underlying the findings in their manuscript fully available?

Reviewer #1: Yes

Reviewer #2: No

Reviewer #3: Yes

4. Is the manuscript presented in an intelligible fashion and written in standard English?

Reviewer #1: Yes

Reviewer #2: Yes

Reviewer #3: Yes

5. Review Comments to the Author

Reviewer #1: The manuscript is to evaluate effects of GLP1R agonists on HbA1c and body weight reductions in real world population by meta-analysis methodology. Authors found positive association between HbA1c and body weight reductions by GLP1R agonists treatment in a diabetes patient population.

I have a couple of critiques authors should address.

1. HbA1c reduction by GLP1R agonists treatment is interacted with the baseline HbA1c. Therefore, restricting population to DM patients skews associations between body weight and HbA1c effects to make another bias. Authors should touch on the issue.

2. Authors do not take adverse effects into account in the analyses. Nausea is well-known adverse effect for GLP1R agonists. Hence, the AE could reduce appetite to modify body weight in entire population. Is there any interaction between AE and body weight?

3. Line 296-321; Authors discuss about genetic polymorphisms as the potential reasons for different HbA1c responses. I do not understand the logic here. What kind of genetic polymorphisms is authors suggesting? The cited paper does not explain any polymorphisms for the GLP1R agonist responses. Is it a bit too haste to discuss about genetic differences for the different HbA1c effects?

4. Line 216-221; Numbers here do not match to the table 2. Please double check.

5. Reference #57 is irrelevant.

6. Table 1; Baseline BMI is not interacted with GLP1R agonists mediated body weight changes. How about to use baseline body weight?

7. Table 2 and S5 Table; What is a definition of tao?

Reviewer #2: With reference to the review of Manuscript number PONE-D-22-25301 “Association between glycemic control and weight reduction of glucagon-like peptide-1 receptor agonists: a multivariate meta-analysis”

Following views are there:

A useful study in a way that obese adults are prone to development of diabetes the study shows that glucagon-like peptide-1 receptor agonists improve glycaemia and decreases the body weight, both effects can protect obese adults from developing diabetes.

However many background con founders can effect the weight gain or loss therefore stratification is desired like patient with and without background Insulin treatment will have different effect on body weight reduction. Different duration of treatment can affect body weight differently. Kindly include non diabetic patients in Table 2 ,5. (All Participants, diabetic and non diabetic participants.)

Quality assessment of primary studies to evaluate the reliability of study results is an essential and mandatory part of meta-analyses. Which scale was used for present study to asses the quality ? Kindly provide score of the studies in table 3.

Adverse events, including nausea, diarrhea, headaches, dizziness and vomiting Associated with glucagon-like peptide-1 receptor agonists can be different in different subsets and will affect eating and weight loss differently. Were there unacceptable side effects in your study, which can outweigh the weight loss benefits in any subset?

Over all a nice study can be published with minor improvements.

Reviewer #3: 1. Title: Ambiguous- the authors need to modify the title. We suggest “Effect of glucagon-like peptide-1 receptor agonists on glycemic control, and weight reduction in adults: a multivariate meta-analysis”

2. Abstract

a) Aims: Requires modification to be in consonance with the new title

b) Methods: First sentence- The authors should state the exact time of commencement of the study and not just state “from study inception”.

c) Conclusion: The last sentence should be moved to the results section. However, the implication(s) of the result contained in the sentence can be included in the conclusion section as done in the conclusion section of the manuscript.

3. Introduction: Well written

4. Materials and Methods:

a) Line 61: There is the need for the authors to define the word “inception” and state categorically the date the study commenced OR did the authors mean since the inception of the databases? If yes, this should be expressly stated.

b) Lines 83 – 85: The last sentence in the methodology section should be moved to appropriate place as it described specific roles of some authors.

5. Data analysis: This section is too long. I would suggest that the authors revise lines 88 – 99 or move it to the supplementary section..

6. Results:

a) Table 1: The authors should unbold the results in bold format. The contents of the Table are already explained in lines 155 – 159 and the p-values are self-explanatory.

b) There is the need to ensure uniform decimal points for the p-values.

c) The authors should remove the sentence “bold indicates statistical significance” from the legend since the results will be unbolded.

d) Going through row 3 in Table 1, Age vs Body weight has p-value of 0.072 which is not significant but was bolded.

7. Discussion:

a) Line 263- Long-acting GLP-1 Ras “showed” and not “show”.

8. Conclusion: Sentence 2- Replace “could” with “might”

Other Comments

1. Inclusion of withdrawn, terminated and unknown status studies as stated in lines 278 – 280 is a source of concern as they could obscure the true findings and conclusion of this study. Is it possible for the authors to reanalyze the retrieved data without the aforementioned studies and find out if a similar outcome will still emerge?

2. Did the authors exclude studies that included type 2 diabetics on insulin in the multivariate analysis as this could affect both the glycemic control and body weight?

3. The observed negative correlation between glycemic control and weight reduction in the univariate analysis is not unexpected because of the diverse population of the participants, different types of GLP-1 RAs and the publication bias.

Suggestion

This manuscript might benefit from a Biostatistician input(s) as I have limited knowledge on the statistical tools/methods used in the study.

6. PLOS authors have the option to publish the peer review history of their article (what does this mean?). If published, this will include your full peer review and any attached files.

Reviewer #1: **Yes: **Yuichiro Adachi

Reviewer #2: No

Reviewer #3: No

---

## [Author Response · Author response to Decision Letter 0]

31 Oct 2022

Please see attached files "Response to Reviewers".

Response to reviewers:

PONE-D-22-25301

Association between glycemic control and weight reduction of glucagon-like peptide-1 receptor agonists: a multivariate meta-analysis

Response: Thanks for your advice. We had changed the file names of the supplements.

Response: Thanks for your advice. We included our amended statements within the cover letter that “The authors received no specific funding for this work.”

3. We note that this manuscript is a systematic review or meta-analysis; our author guidelines therefore require that you use PRISMA guidance to help improve reporting quality of this type of study. Please upload copies of the completed PRISMA checklist as Supporting Information with a file name “PRISMA checklist” and it should be uploaded separately.

Response: Thanks for your advice. We upload copies of the completed PRISMA checklist as Supporting Information with a file name “PRISMA checklist” and it was uploaded separately.

Reviewers' comments:

Comments to the Author

1. Is the manuscript technically sound, and do the data support the conclusions?

Reviewer #1: Yes

Reviewer #2: Partly

Reviewer #3: Yes

2. Has the statistical analysis been performed appropriately and rigorously?

Reviewer #1: Yes

Reviewer #2: Yes

Reviewer #3: I Don't Know

3. Have the authors made all data underlying the findings in their manuscript fully available?

Reviewer #1: Yes

Reviewer #2: No

Reviewer #3: Yes

4. Is the manuscript presented in an intelligible fashion and written in standard English?

Reviewer #1: Yes

Reviewer #2: Yes

Reviewer #3: Yes

5. Review Comments to the Author

Reviewer #1: 

The manuscript is to evaluate effects of GLP1R agonists on HbA1c and body weight reductions in real world population by meta-analysis methodology. Authors found positive association between HbA1c and body weight reductions by GLP1R agonists treatment in a diabetes patient population. I have a couple of critiques authors should address.

1. HbA1c reduction by GLP1R agonists treatment is interacted with the baseline HbA1c. Therefore, restricting population to DM patients skews associations between body weight and HbA1c effects to make another bias. Authors should touch on the issue.

Response: Thanks for your advice. We revised the discussion to touch on the issue: However, marked weight loss in a non-diabetic patient in response to GLP-1 RA treatment did not indicate that clinicians could expect a corresponding glycemic improvement due to the results were interacted with the underlying HbA1c level genetic polymorphisms.

2. Authors do not take adverse effects into account in the analyses. Nausea is well-known adverse effect for GLP1R agonists. Hence, the AE could reduce appetite to modify body weight in entire population. Is there any interaction between AE and body weight? 

Response: Thanks for your advice. Considering GI side effect could reduce appetite to modify body weight in entire population, we performed the interaction between AE and outcomes and the results were shown in Table 1.

Revised results: Meta-regression (Table 1) showed that the pooled HbA1c reduction significantly interacted with participants’ baseline age (p = 0.032), …, and gastrointestinal side effect (p = 0.002) but did not interact with ….. ; a 1% increase in the proportion of female participants significantly increased the pooled HbA1c change by 0.015%; …. and 1% increase in gastrointestinal side effect significantly increase the pooled HbA1c level by 0.017%.…………….

Meta-regression (Table 1) showed that the pooled body weight reduction significantly interacted with the proportion of female participants (p = 0.016),…and gastrointestinal side effect (p = 0.025), and ... A 1% increase in the proportion of female participants significantly decreased the pooled body weight reduction by 0.065 kg; …..; a 1% increase in gastrointestinal side effect significantly decreased the pooled body weight reduction by 0.059 kg. 

3. Line 296-321; Authors discuss about genetic polymorphisms as the potential reasons for different HbA1c responses. I do not understand the logic here. What kind of genetic polymorphisms is authors suggesting? The cited paper does not explain any polymorphisms for the GLP1R agonist responses. Is it a bit too haste to discuss about genetic differences for the different HbA1c effects?

Response: Thanks for your advice. We deleted the sentences. The revised discussion: The mechanism underlying the variable effects of GLP-1 RAs on body weight is not well understood. GLP-1 RAs decreased appetite through direct effects on the hypothalamus, neuronal activation in brain areas, reduced caloric intake, and interference of effective compensatory mechanisms counteracting weight loss.[60, 61].[1, 2] Genetic polymorphisms through signal transduction pathways from different hypothalamic effects in GLP-1 RA responders and non-responders[63] partially explained this difference.

….

Lin 304: However, marked weight loss in a non-diabetic patient in response to GLP-1 RA treatment did not indicate that clinicians could expect a corresponding glycemic improvement due to the results were interacted with the underlying HbA1c level genetic polymorphisms.

4. Line 216-221; Numbers here do not match to the table 2. Please double check.

Response: Thanks for your advice. We matched the numbers in the results with the table 2. The revised results:

Results of the structural equation modeling multivariate meta-analysis

Maximum likelihood estimation worked well in the analysis. Table 2 shows that the pooled HbA1c change induced by GLP1-RAs was -0.85% (95% CI [-1.03%, -0.66%], I2 = 99%), and the pooled body weight change was -4.03 kg (95% CI [-5.11 kg, -2.95 kg], I2 = 99%), which were similar to the results of the univariate meta-analysis. However, overall, the pooled between-study level correlation coefficient between HbA1c and body weight changes from baseline was -0.42, which was the opposite of the within-study level. To explore the negative correlation, we further restricted the multivariate analysis to participants with diabetes. The pooled HbA1c change by GLP1-RA was -0.96% (95% CI [-1.14%, -0.79%], I2 = 96%), the pooled body weight change was -3.23 kg (95% CI [-3.86 kg, -2.59 kg], I2 = 95%); the amount of between-study heterogeneity of body weight decreased from 7.36 to 1.77 and the 95% CI became narrower. The pooled correlation coefficient turned to a positive estimate of 0.32. And if we restricted the multivariate analysis to participants without diabetes of only five studies. The pooled HbA1c change by GLP1-RA was -0.27% (95% CI [-0.31%, -0.23%], I2 = 13%), the pooled body weight change was -6.76 kg (95% CI [-10.81 kg, -2.72 kg], I2 = 99%); the amount of between-study heterogeneity of body weight was much increased to 20.84 with a wide 95% CI due to the limited included articles. However, the pooled correlation coefficient was positive estimate of 0.81. 

5. Reference #57 is irrelevant.

Response: Thanks for your advice. We deleted the reference #57

Thus, meta-analyses published before 2015….., while more recent meta-analyses usually targeted semaglutide[3-5] or focused on emerging outcomes such as cardiovascular or kidney disease.[6, 7] In contrast, our study aimed to investigate the glycemic control and weight reduction caused by long-acting GLP-1 RAs. Previous studies showed high heterogeneity (I2 = 80%-90%) even for findings related to the same GLP-1 RAs. [3-5] All the potential effect modifiers in our study showed an opposite direction of interaction between glycemic control and weight reduction. Previous meta-analyses have rarely reported this topic and yielded inconsistent results[3-5];

Reference 57. Li J, He K, Ge J, Li C, Jing Z. Efficacy and safety of the glucagon-like peptide-1 receptor agonist oral semaglutide in patients with type 2 diabetes mellitus: A systematic review and meta-analysis. Diabetes research and clinical practice. 2021;172:108656. Epub 2021/01/13. doi: 10.1016/j.diabres.2021.108656. PubMed PMID: 33434602.

6. Table 1; Baseline BMI is not interacted with GLP1R agonists mediated body weight changes. How about to use baseline body weight?

Response: Thanks for your advice. We performed meta-regression with baseline body weight with the outcomes…

Revised results: 

Meta-regression (Table 1) showed that the pooled HbA1c reduction significantly interacted with participants’ baseline age (p = 0.032), proportion of female participants (p = 0.017), the baseline HbA1c level (p = 0.018), and gastrointestinal side effect (p = 0.002) but did not interact with baseline body weight or BMI level, duration of diabetes, follow-up period or insulin use. 

Meta-regression (Table 1) showed that the pooled body weight reduction significantly interacted with the proportion of female participants (p = 0.016), the follow-up period (p = 0.018) and gastrointestinal side effect (p = 0.025), and had a borderline interaction with participants’ baseline age (p = 0.091) and the participants’ baseline HbA1c level (p = 0.077), but did not interact with baseline BMI level or body weight, insulin use, or the duration of diabetes. 

7. Table 2 and S5 Table; What is a definition of tao?

Response: Thanks for your advice. We revised the footnotes of the tables of table 2 and S5 table.

tao, the variance of effect measure

Reviewer #2: 

With reference to the review of Manuscript number PONE-D-22-25301 “Association between glycemic control and weight reduction of glucagon-like peptide-1 receptor agonists: a multivariate meta-analysis”

Following views are there:

A useful study in a way that obese adults are prone to development of diabetes the study shows that glucagon-like peptide-1 receptor agonists improve glycaemia and decreases the body weight, both effects can protect obese adults from developing diabetes.

However many background con founders can effect the weight gain or loss therefore stratification is desired like patient with and without background Insulin treatment will have different effect on body weight reduction. Different duration of treatment can affect body weight differently. Kindly include non diabetic patients in Table 2 ,5. (All Participants, diabetic and non diabetic participants.)

Response: Thanks for your advice, we performed further analyses to all participants, patients with diabetes and participants without diabetes.

Revised results: Table 2 shows that the pooled HbA1c change induced by GLP1-RAs was…. To explore the negative correlation, we further restricted the multivariate analysis to participants with diabetes. …. There were only five studies focused on participants without diabetes. The pooled HbA1c change by GLP1-RA was -0.27% (95% CI [-0.31%, -0.23%], I2 = 13%), the pooled body weight change was -6.76 kg (95% CI [-10.81 kg, -2.72 kg], I2 = 99%); the amount of between-study heterogeneity of body weight was much increased to 20.84 with a wide 95% CI due to the limited included articles. However, the pooled correlation coefficient was positive of 0.81. The pooled results for all participants and the results restricted to patients with diabetes are shown in Fig 3.

Quality assessment of primary studies to evaluate the reliability of study results is an essential and mandatory part of meta-analyses. Which scale was used for present study to asses the quality ? Kindly provide score of the studies in table 3.

Response: Thanks for your advice. In line 83: All included trials were assessed for bias using the Cochrane risk-of-bias tool 2.0.[14] 

Reference #14. Higgins JP, Thomas J, Chandler J, Cumpston M, Li T, Page MJ. Cochrane Handbook for Systematic Reviews of Interventions version 6.2: Cochrane; 2021.

We put the long table with 31 reference included articles in the S3 Table.

Reviewer #3:

1. Title: Ambiguous- the authors need to modify the title. We suggest “Effect of glucagon-like peptide-1 receptor agonists on glycemic control, and weight reduction in adults: a multivariate meta-analysis”

Response: Thanks for your advice. We modified the title to “Effect of glucagon-like peptide-1 receptor agonists on glycemic control, and weight reduction in adults: a multivariate meta-analysis”.

2. Abstract

a) Aims: Requires modification to be in consonance with the new title

Response: Thanks for your advice. We modified the aims of the abstract:

Previous aims of the abstract: To explore the effect of glucagon-like peptide-1 receptor agonist (GLP-1 Ras) on glycemic control and weight reduction in adults.

Revised aims of the abstract: To explore the effect of glucagon-like peptide-1 receptor agonist (GLP-1 Ras) on glycemic control and weight reduction in adults.

b) Methods: First sentence- The authors should state the exact time of commencement of the study and not just state “from study inception”.

Response: Thanks for your advice. We modified the methods of the abstract: Databases were searched from study inception August 2021 to March 2022.

c) Conclusion: The last sentence should be moved to the results section. However, the implication(s) of the result contained in the sentence can be included in the conclusion section as done in the conclusion section of the manuscript.

Response: Thanks for your advice. The last sentence of the manuscript was moved to the results section.

Original abstract:

Results. ….The standardized pooled correlation coefficient between HbA1c levels and body weight was -0.40.

Conclusion. Long-acting GLP-1 RAs significantly reduced the glycated hemoglobin level and body weight in adults. However, a negative correlation between glycemic control and weight reduction was obtained.

Revised abstract:

Results. ….The standardized pooled correlation coefficient between HbA1c levels and body weight was -0.40. A negative correlation between glycemic control and weight reduction was obtained.

Conclusion. Long-acting GLP-1 RAs significantly reduced the glycated hemoglobin level and body weight in adults. However, a negative correlation between glycemic control and weight reduction was obtained.

Original conclusion of the manuscript:

In conclusion, long-acting GLP-1 RAs significantly lowered HbA1c levels and body weight in adults. However, the positive association between glycemic control and weight reduction in diabetic patients could not be expected in non-diabetic patients treated with long-acting GLP-1 RAs.

Revised conclusion of the manuscript:

In conclusion, long-acting GLP-1 RAs significantly lowered HbA1c levels and body weight in adults. However, the positive association between glycemic control and weight reduction was only observed in diabetic patients and in non-diabetic participants, but not in all participants with high heterogeneity treated with long-acting GLP-1 RAs.

3. Introduction: Well written

4. Materials and Methods:

a) Line 61: There is the need for the authors to define the word “inception” and state categorically the date the study commenced OR did the authors mean since the inception of the databases? If yes, this should be expressly stated.

Response: Thanks for your advice. We revised the methos:

Original Search strategy and selection criteria

We searched the Medline, Ovid EMBASE, Cochrane Library and ClinicalTrials.gov databases for relevant studies from inception to March 2022 by using the following keywords:

Revised Search strategy and selection criteria

We searched the Medline, Ovid EMBASE, Cochrane Library and ClinicalTrials.gov databases for relevant studies from inception August 2021 to March 2022 by using the following keywords:

b) Lines 83 – 85: The last sentence in the methodology section should be moved to appropriate place as it described specific roles of some authors.

Response: Thanks for your advice. We deleted the last sentence in the methodology section: All included trials were assessed for bias using the Cochrane risk-of-bias tool 2.0.[14] The authors M. C. T. and W. H. T. conducted the searches and conducted quality assessments independently, and disagreements were resolved through consensus.

5. Data analysis: This section is too long. I would suggest that the authors revise lines 88 – 99 or move it to the supplementary section..

Response: Thanks for your advice. We move lines 88 – 99 to the supplementary section.

Revised methods:…All included trials were assessed for bias using the Cochrane risk-of-bias tool 2.0.[14]. The details of the data extraction in our study were described in supplement (S1 File).

S1 File. Details of data extraction in the study: The outcomes corresponding to glycemic changes were usually differences in HbA1c levels, changes in fasting plasma glucose levels, changes in self-monitored blood glucose levels from baseline, differences between groups, rate of achievement of an HbA1c target of 6.5% or 7.0%, or other measurements of insulin level or homeostatic model assessment scores among different studies. The outcomes of anthropometric changes were usually differences in body weight, body mass index (BMI), or waist circumference from baseline, differences between groups, or the rate of achievement of a 5%–10% reduction in body weight or BMI. To avoid unit-of-analysis errors,[15] we only extracted the most commonly reported, estimated treatment difference in HbA1c level and body weight from the baseline corresponding to a full dose of GLP-1 RAs in each RCT. The estimated treatment difference was determined using the Revman calculator with the numbers in each arm and the p-values.

6. Results:

a) Table 1: The authors should unbold the results in bold format. The contents of the Table are already explained in lines 155 – 159 and the p-values are self-explanatory.

Response: Thanks for your advice. We unbold the results in bold format.

b) There is the need to ensure uniform decimal points for the p-values.

Response: Thanks for your advice. We uniformed the decimal points for the p-values.

c) The authors should remove the sentence “bold indicates statistical significance” from the legend since the results will be unbolded.

Response: Thanks for your advice. We remove the sentence “bold indicates statistical significance” from the legend

d) Going through row 3 in Table 1, Age vs Body weight has p-value of 0.072 which is not significant but was bolded.

Response: Thanks for your advice. We unbold the results in Tables now.

7. Discussion:

a) Line 263- Long-acting GLP-1 Ras “showed” and not “show”.

Response: Thanks for your advice. We revised the sentence: Long-acting GLP-1 RAs showed better efficacy in weight reduction and glycemic control than short-acting GLP-1 RAs.[39]

8. Conclusion: Sentence 2- Replace “could” with “might”

Response: Thanks for your advice. We revised the discussion: Our meta-analysis demonstrated that long-acting GLP-1 RAs significantly reduced HbA1c levels and body weight in adults. The high heterogeneity in our study could might be attributed to the different GLP-1 RAs and the diverse populations ranging from non-diabetic overweight/obese participants to patients with diabetes complicated with end organ damage.

Other Comments

1. Inclusion of withdrawn, terminated and unknown status studies as stated in lines 278 – 280 is a source of concern as they could obscure the true findings and conclusion of this study. Is it possible for the authors to reanalyze the retrieved data without the aforementioned studies and find out if a similar outcome will still emerge?

Response: Thanks for your advice. We did not analyze the articles in lines 278 – 280, the withdrawn, terminated and unknown status studies. We mentioned the above studies in the discussion due to these studies are potential unpublished studies, the possible source of the publication bias.

2. Did the authors exclude studies that included type 2 diabetics on insulin in the multivariate analysis as this could affect both the glycemic control and body weight?

Response: Thanks for your advice. We did not exclude studies that included type 2 diabetics on insulin in the multivariate analysis, thus insulin could affect both the glycemic control and body weight. We performed meta-regression with insulin with the outcomes: 

The revised results:

Meta-regression (Table 1) showed that the pooled HbA1c reduction significantly interacted with participants’ baseline age (p = 0.032), proportion of female participants (p = 0.017), the baseline HbA1c level (p = 0.018), and gastrointestinal side effect (p = 0.002) but did not interact with baseline body weight or BMI level, duration of diabetes, follow-up period or insulin use….

Meta-regression (Table 1) showed that the pooled body weight reduction significantly interacted with the proportion of female participants (p = 0.016), the follow-up period (p = 0.018) and gastrointestinal side effect (p = 0.025), and had a borderline interaction with participants’ baseline age (p = 0.091) and the participants’ baseline HbA1c level (p = 0.077), but did not interact with baseline BMI level or body weight, insulin use, or the duration of diabetes….

3. The observed negative correlation between glycemic control and weight reduction in the univariate analysis is not unexpected because of the diverse population of the participants, different types of GLP-1 RAs and the publication bias.

Suggestion

This manuscript might benefit from a Biostatistician input(s) as I have limited knowledge on the statistical tools/methods used in the study.

Response: Thanks for your advice. The manuscript was supervised under a chief biostatistician professor in Institute of Epidemiology and Preventive Medicine, College of Public Health, National Taiwan University, Taipei, Taiwan

6. PLOS authors have the option to publish the peer review history of their article (what does this mean?). If published, this will include your full peer review and any attached files.

Do you want your identity to be public for this peer review? For information about this choice, including consent withdrawal, please see our Privacy Policy.

Reviewer #1: Yes: Yuichiro Adachi

Reviewer #2: No

Reviewer #3: No

---

## [Decision Letter · Decision Letter 1]

22 Nov 2022

Effect of glucagon-like peptide-1 receptor agonists on glycemic control, and weight reduction in adults: a multivariate meta-analysis

PONE-D-22-25301R1

Dear Dr. Yeh,

We’re pleased to inform you that your manuscript has been judged scientifically suitable for publication and will be formally accepted for publication once it meets all outstanding technical requirements.

Kind regards,

Ming-Chang Chiang

Academic Editor

PLOS ONE

Additional Editor Comments (optional):

Reviewers' comments:

Reviewer's Responses to Questions

**Comments to the Author**

1. If the authors have adequately addressed your comments raised in a previous round of review and you feel that this manuscript is now acceptable for publication, you may indicate that here to bypass the “Comments to the Author” section, enter your conflict of interest statement in the “Confidential to Editor” section, and submit your "Accept" recommendation.

Reviewer #1: All comments have been addressed

Reviewer #2: All comments have been addressed

2. Is the manuscript technically sound, and do the data support the conclusions?

Reviewer #1: Yes

Reviewer #2: Yes

3. Has the statistical analysis been performed appropriately and rigorously? 

Reviewer #1: Yes

Reviewer #2: Yes

4. Have the authors made all data underlying the findings in their manuscript fully available?

Reviewer #1: Yes

Reviewer #2: Yes

5. Is the manuscript presented in an intelligible fashion and written in standard English?

Reviewer #1: Yes

Reviewer #2: Yes

6. Review Comments to the Author

Reviewer #1: I congratulate authors to address all comments appropriately. Now, the manuscript is a good shape to publish.

I found a typo at line 225 (S5 Table). Please correct before publication.

Reviewer #2: May be accepted with current revisions. All comments were addressed. Quality has definitely improved. Will be a useful paper and guide further research.

7. PLOS authors have the option to publish the peer review history of their article (what does this mean?). If published, this will include your full peer review and any attached files.

Reviewer #1: **Yes: **Yuichiro Adachi

Reviewer #2: No

---

## [Editor Report · Acceptance letter]

24 Nov 2022

PONE-D-22-25301R1 

Effect of glucagon-like peptide-1 receptor agonists on glycemic control, and weight reduction in adults: a multivariate meta-analysis 

Dear Dr. Yeh:

I'm pleased to inform you that your manuscript has been deemed suitable for publication in PLOS ONE. Congratulations! Your manuscript is now with our production department. 

Kind regards, 

on behalf of

Dr. Ming-Chang Chiang 

Academic Editor

PLOS ONE